# Temporal Coherent Test-Time Optimization for Robust Video Classification

**Chenyu Yi**[1]    **Siyuan Yang**[1,2]    **Yufei Wang** [1]    **Haoliang Li**[3*]    **Yap-Peng Tan**[1]    **Alex C. Kot**[1]

[1]School of Electrical and Electronic Engineering, Nanyang Technological University, Singapore
[2]Interdisciplinary Graduate Programme, Nanyang Technological University, Singapore
[3]Department of Electrical Engineering, City University of Hong Kong, China
{yich0003,siyuan005,yufei001}@e.ntu.edu.sg    haoliang.li@cityu.edu.hk    {eyptan,eackot}@ntu.edu.sg

## Abstract

Deep neural networks are likely to fail when the test data is corrupted in real-world deployment (e.g., blur, weather, etc.). Test-time optimization is an effective way that adapts models to generalize to corrupted data during testing, which has been shown in the image domain. However, the techniques for improving video classification corruption robustness remain few. In this work, we propose a **Te**mporal **Co**herent Test-time Optimization framework (TeCo) to utilize spatio-temporal information in test-time optimization for robust video classification. To exploit information in video with self-supervised learning, TeCo minimizes the entropy of the prediction based on the global content from video clips. Meanwhile, it also feeds local content to regularize the temporal coherence at the feature level. TeCo retains the generalization ability of various video classification models and achieves significant improvements in corruption robustness across Mini Kinetics-C and Mini SSV2-C. Furthermore, TeCo sets a new baseline in video classification corruption robustness via test-time optimization.

## 1    Introduction

Deep neural networks have achieved tremendous success in many computer vision tasks when the training and test data are identically and independently distributed (i.i.d). However, the mismatch between them is common when the model is deployed in the real world (Wang et al. (2022b); Li et al. (2020)). For example, weather changes like rain and fog, and data pre-processing like saturate adjustment and compression can corrupt the test data. Many works show that the common corruptions arising in nature can degrade the performance of models at test time significantly (Hendrycks & Dietterich, 2019; Yi et al., 2021; Kar et al., 2022; Geirhos et al., 2018; Yu et al., 2022). In video classification, Yi et al. (2021) demonstrates the vulnerability of models against corruptions like noise, blur, and weather variations.

Various techniques have been proposed to improve the robustness of models against common corruptions (a.k.a.corruption robustness). The most popular direction is increasing the diversity of input training data via data augmentations and applying regularization at training time (Zheng et al., 2016; Wang et al., 2021b; Hendrycks et al., 2020; Rusak et al., 2020; Wang et al., 2020). It trains one model and evaluates on all types of corruption. However, the training data is often unavailable because of privacy concerns and policy constraints, making it more challenging to deploy models in the real world. In this work, we focus on the direction of test-time optimization. Under such a scheme, the parameters of one model will be optimized by one type of corrupted test data specifically. Test-time optimization updates the model to fit into the deployment environment at test time, without access to training data. There are several test-time optimization techniques emerging in image-based tasks (Wang et al., 2021a; Schneider et al., 2020; Liang et al., 2020; Sun et al., 2020). However, we find these techniques are not able to generalize to video-based corruption robustness tasks well from empirical analysis. We hypothesize the gap between image and video-based tasks comes from several aspects. Firstly, the corruptions in the video can change temporal information,

---

*Corresponding Author

which requires the model to be both generalizable and robust (Yi et al., 2021). Hence, improving model robustness against the corruptions in video like bit error, and frame rate conversion is more challenging. Secondly, video input data has a different format from image data. For example, video data has a much larger size than image data. It is impractical to use a similar batch size as image-based tasks (e.g., batch size of 256 in Tent (Wang et al., 2021a)), though the batch size is an important hyper-parameter in test-time optimization (Wang et al., 2021a; Schneider et al., 2020). Lastly, these techniques ignore the huge information hidden in the temporal dimension.

To improve the video classification model robustness against corruptions consistently, we propose a temporal coherent test-time optimization framework **TeCo**. TeCo is a test-time optimization technique with two self-supervised objectives. We propose to build our method upon the test-time optimization which updates all the parameters in shallow layers and only normalization layers in the deep layers. It can benefit from both training and test data, and such an optimization strategy remains part of model parameters and statistics obtained at training time and updates the unfrozen parameters with test information. Besides, we utilize global and local spatio-temporal information for self-supervision. We use uniform sampling to ensure global information in input video data and optimize the model parameters via entropy minimization. By dense sampling, we extract another local stream that has a smaller time gap between consecutive frames. Due to the smooth and continuous nature of adjacent frames in the video (Li & DiCarlo, 2008; Wood & Wood, 2016), we apply a temporal coherence regularization as a self-supervisor in test-time optimization on the local pathway. As such, our proposed technique enables the model to learn more invariant features against corruption. As a result, TeCo achieves promising robustness improvement on two large-scale video corruption robustness datasets, Mini Kinetics-C and Mini SSV2-C. Its performance is superior across various video classification backbones. TeCo increases average accuracy by 6.5% across backbones on Mini Kinetics-C and by 4.1% on Mini SSV2-C, which is better than the baseline methods Tent (1.9% and 0.9%) and SHOT (1.9% and 2.0%). Additionally, We show that TeCo can guarantee the smoothness between consequent frames at the feature level, which indicates the effectiveness of temporal coherence regularization.

We summarize our contributions as follows:

- To the best of our knowledge, we make the first attempt to study the test-time optimization techniques for video classification corruption robustness across datasets and model architectures.
- We propose a novel test-time optimization framework TeCo for video classification, which utilizes spatio-temporal information in training and test data to improve corruption robustness.
- For video corruption robustness, TeCo outperforms other baseline test-time optimization techniques significantly and consistently on Mini Kinetics-C and Mini SSV2-C datasets.

## 2 RELATED WORKS

### 2.1 PROBLEM SETTING

**Corruption Robustness.** Deep neural networks are vulnerable to common corruptions generated in real-world deployment (Geirhos et al., 2017; Hendrycks & Dietterich, 2019). Hendrycks & Dietterich (2019) firstly propose ImageNet-C to benchmark the common corruption robustness of deep learning models. It assumes the tested model has no prior knowledge of the corruption arising during test time. The model is trained with clean data while tested on corrupted data. Under such a setting, we are able to estimate the overall robustness of models against corruption. In the following, benchmark studies and techniques across various computer vision tasks are booming (Kar et al., 2022; Michaelis et al., 2019; Kamann & Rother, 2020). These studies on corruption robustness bridge the gap between research in well-setup lab environments and deployment in the field. Recently, studies have emerged on corruption robustness in video classification (Yi et al., 2021; Wu & Kwiatkowska, 2020). In this work, we tap the potential of spatio-temporal information in video data and improve the corruption robustness of video classification models during testing.

**Video Classification.** Recently, with the introduction of a number of large-scale video datasets such as Kinetics (Carreira & Zisserman, 2017), Something-Something V2 (SSV2) (Goyal et al.,

2017), and Sports1M (Karpathy et al., 2014), video classification has attracted increasing attention. Most existing video classification works mainly focus on two perspectives: improving accuracy (Simonyan & Zisserman, 2014; Wang et al., 2016; Carreira & Zisserman, 2017; Hara et al., 2017; Xie et al., 2018; Yang et al., 2020; Bertasius et al., 2021; Li et al., 2022) and model efficiency (Feichtenhofer et al., 2019; Lin et al., 2019; Feichtenhofer, 2020; Kondratyuk et al., 2021; Wang et al., 2021c). However, as mentioned above, techniques for improving video classification corruption robustness remain few. In this paper, we focus on the optimization of video corruption robustness and conduct experiments on Mini Kinetics-C and Mini SSV2-C datasets. The Kinetics dataset relies on spatial semantic information for video classification, while SSV2 contains more temporal information. It enables us to evaluate the effectiveness of our proposed optimization method on these two different types of video data.

## 2.2 TEST-TIME OPTIMIZATION

Optimizing the model with unlabeled test data has been used in many machine learning tasks (Shu et al., 2022; Huang et al., 2020; Shocher et al., 2018; Azimi et al., 2022; Niu et al., 2022; Wang et al., 2022a). These methods only require the pretrained model and test data for optimization in the inference. Test-Time Training (TTT) (Sun et al., 2020) and TTT++ (Liu et al., 2021) update models at test-time, but they jointly optimize the supervised loss and the self-supervised auxiliary loss in training. Though these methods demonstrate promising improvement in image-based corruption robustness, there is little progress made in video-based test-time optimization. We make the first attempt to utilize the temporal information for model robustness under the test-time optimization setting.

## 2.3 TEMPORAL COHERENCE IN VIDEO

The concept of temporal coherence makes an assumption that adjacent frames in video data have semantically similar information (Goroshin et al., 2015). The assumption is based on the phenomenon that the visual world is smoothly varying and continuous. The stability of the visual world in the temporal dimension is significant for many studies in biological vision (Li & DiCarlo, 2008; Wood, 2016; Wood & Wood, 2016). With the assumption, the invariance between neighboring frames in the video can provide self-supervision for many computer vision tasks (Jayaraman & Grauman, 2015; Wiskott & Sejnowski, 2002; Dwibedi et al., 2019; Wang et al., 2019). For example, Jayaraman & Grauman (2015) exploits motion signal in the egocentric video to regularize image recognition task; Wang et al. (2019) utilizes the cycle consistency in time to learn visual temporal correspondence. In the corruption robustness problem, we use temporal coherence as a self-supervisor to regularize models to be more invariant against corruption.

## 3 METHOD

We propose an end-to-end training framework to improve the robustness of video classification models against corruption at test time. In such a framework, we optimize pre-trained deep learning models during the test time and minimize two objectives without label information. These two objectives correspond to two pathways. The first pathway uses a global stream as input and minimizes the entropy of prediction; the second pathway leverages a local stream as input and regularizes the temporal coherence among consecutive video frames. We integrate these two pathways and named our framework as **TeCo**. Figure 1 outlines our temporal coherent test-time optimization framework. The backbone in Figure 1 is initialized by a pre-trained model. Any deep video classification model pretrained under a supervised learning scheme can fit into it.

## 3.1 TEST-TIME OPTIMIZATION STRATEGY

Though BN (Schneider et al., 2020) and Tent (Wang et al., 2021a) achieve promising performance on image-based test-time optimization tasks, they only show minor improvements on video data and sometimes degrade the robustness on corruptions in our empirical analysis. BN only adapts the mean and variance of normalization layers by test data partially, it limits the searching space of optimization. Tent removes the training time batch normalization statistics fully, while video classification tasks usually have smaller batch size. It can only capture sub-optimal statistics when just relying on test-time data.

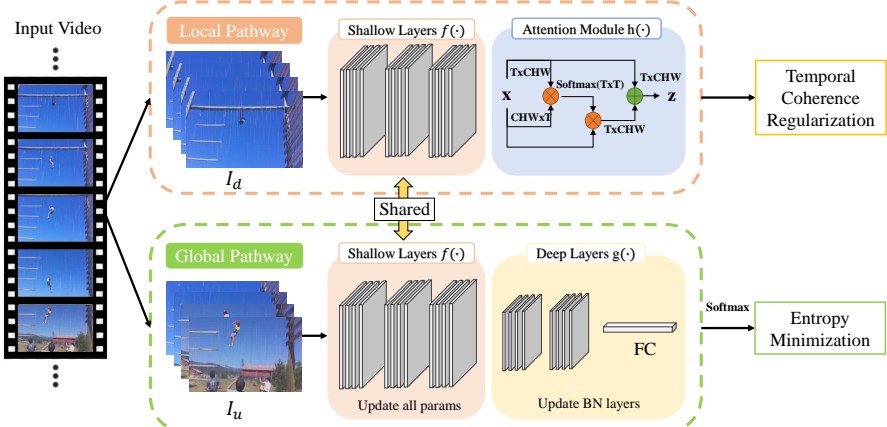

**Figure 1:** The proposed framework TeCo at the test-time optimization stage. TeCo consists of two pathways: the local pathway uses dense video stream $I_d$ as input. It passes through the shallow layers $f$ and attention module $h$ to generate an attention-based feature map $z$. Then it applies temporal coherence regularization on the feature map. We update all the parameters in the shallow layers in this pathway; the global pathway uses a sparse video stream $I_u$, which captures the long-term information. TeCo minimizes the prediction entropy in the global pathway by updating all the parameters in the shallow layers and only normalization parameters in the deep layers.

We perform a simple but effective mechanism by grafting the advantage of these two classic techniques. Following (Schneider et al., 2020), we initialize mean $\mu = \alpha\mu_s + (1 - \alpha)\mu_t$ and variance $\sigma^2 = \alpha\sigma_s^2 + (1-\alpha)\sigma_t^2$, where $s$ stands for training, and $t$ stands for testing. $\alpha$ is the hyper-parameter that controls the weights of training and test time statistics. At the same time, we update the affine transformation parameters in the normalization layers, which helps the model utilize both training and test-time statistics after adaptation.

Apart from normalization layers, we also fine-tune other network parameters (e.g., Conv Layers) in the shallow layers, which are the initial (lower) layers in the network. The shallow layers of robust models generate smoother feature tensors that are better aligned with human perception (Xie et al., 2019; Tsipras et al., 2018). They play the role of denoiser when encountering natural corruptions arising in input data. Inspired by this phenomenon, we update all the parameters in the shallow layers and only normalization layers in the deep layers, as shown in Figure 1. It gives more flexibility to the shallow layers while not interrupting the higher-level semantic information. The effect of different divisions of the shallow and deep layers is explored in the ablation study.

## 3.2 GLOBAL AND LOCAL PATHWAYS

In the test-time optimization of TeCo, we use two pathways (i.e, global pathway and local pathway) and two objectives to optimize model. The global pathway uses uniform sampling to extract frames from the input video $I_u$. The uniform sampling splits the video into multiple clips with equal length. Then we randomly take one frame from each clip. As a result, the selected frames by uniform sampling can capture the global information effectively (Chen et al., 2021). In this pathway, we optimize the overall parameters of shallow layers and the transformation parameters in deep normalization layers by minimizing the entropy of prediction. The objective is:

$$L_{ent} = H(Softmax(g(f(I_u)))) \tag{1}$$

where $g(\cdot)$ represents the deep layers, $f(\cdot)$ stands for the shallow layers, $H$ is the Shannon entropy of prediction $\hat{y}$ with N classes: $H(\hat{y}) = -\sum_{n=1}^{N} p(\hat{y}_n) \log p(\hat{y}_n)$. The Shannon entropy serves as the optimization of the global pathway.

The local pathway leverages dense sampling to take consecutive frames $I_d$, at a random location of the input video, which randomly chooses a location as the center of the sampled sequence. Compared to the large time gap in frames by uniform sampling, the time gap in the frames by dense sampling can be as low as one. In this pathway, the temporal coherence between frames is much stronger. Because the visual world is smoothly varying, the neighboring frames in video data will not change

abruptly in the temporal dimension. When common corruptions interrupt the natural structure of video data, we apply a temporal-coherent regularization to the output feature of the shallow layer. The regularization can be achieved by enforcing the feature similarity in consecutive feature maps of corrupted data. We explicitly represent it with the loss function: $L_{coherence} = \sum_t ||x^t - x^{t-i}||_1$. We use $l_1$ distance to measure the similarity of feature tensors since it penalizes more on outliers and benefits the robustness (Alizadeh et al., 2020; Bektaş & Şişman, 2010). $x^t$ is the tensor at time $t$, which is the output after $f(I_d^t)$. $i$ is the time gap between two neighboring tensors. $L_{coherence}$ encourages the layers to generate smoother features. Correspondingly, the feature extractor will be more invariant to the perturbation, corruption, and disruption in the input data. It leads to more robust video classification models.

## 3.3 ATTENTIVE TEMPORAL COHERENCE REGULARIZATION

When we apply $L_{coherence}$ on feature tenors, we simply assume the frames are varying in a uniform speed because of the same weights along the time dimensions. However, the changes in neighboring frames have different scales in the real world. The example of bungee jumping in Figure 1 shows that the change is small at the beginning of the video because the background is fixed. It has only the blue sky and a jumping platform. When the person is reaching the ground, the background changes rapidly, and there are houses and other people appearing. Hence, we propose to assign different weights to different frames. To encourage temporal consistency in the static background, the slower changes in temporal dimension correspond to higher weights in regularization. The weights can be generated by an attention module $h(\cdot)$. In this work, we use the time dimension-only 1D non-local operation (Wang et al., 2018) to obtain the weights along the time dimension. Hence, the attentive temporal coherent regularization can be written as:

$$L_{att-co} = \sum_t ||h(f(I_d))^t - h(f(I_d))^{t-i}||_1, \qquad (2)$$

where $h(\cdot)$ is the attention module.

## 3.4 OVERALL OBJECTIVE FUNCTION

To summarize, the global pathway uses $I_u$ as input and entropy minimization as objective, and the local pathway leverages $I_d$ as input and attentive temporal coherence regularization as objective. The overall objective function for TeCo can be represented as:

$$L = L_{ent}(I_u, f, g) + \beta L_{att-co}(I_d, f, h), \qquad (3)$$

where $\beta > 0$ is a balancing hyper-parameter.

## 4 EXPERIMENTS

We evaluate the corruption robustness of TeCo on Mini Kinetics-C and Mini SSV2-C. The mean performance on corruption is the main metric for robustness measurement. We also compare it with other baseline methods across architectures.

**Dataset.** Kinetics (Carreira & Zisserman, 2017) and Something-Something-V2 (SSV2) (Goyal et al., 2017) are two most popular large-scale datasets in video classification community. Kinetics is extracted from the Youtube website and it relies on spatial information for classification. As complementary, SSV2 is a first-view video dataset constructed systematically in the lab environment, which has more temporal changes. We use their variants Mini Kinetic-C and Mini SSV2-C (Yi et al., 2021) to evaluate the robustness of models against corruptions. The mini-version datasets randomly sample half of the classes from the original large-scale datasets. Hence, Mini Kinetics-C contains around 10K videos in the test dataset; Mini SSV2-C has a test dataset with a size of around 13K videos. These two datasets apply 12 types of corruptions arising in nature on the original clean test datasets. Each corruption has 5 levels of severities. The samples of Mini Kinetics-C and Mini SSV2-s are shown in supplementary.

**Metrics.** The classification accuracy on corrupted data is the most common metric to measure the corruption robustness of models. In our experiments, the corrupted datasets have 12 types of

corruptions and each has 5 levels of severities. We use $mPC$ (stands for mean performance on corruptions) to evaluate the overall robustness. We compute $mPC$ by averaging the classification accuracy as: $mPC = \sum_{c=1}^{12} \sum_{s=1}^{5} CA_{c,s}$, where $CA_{c,s}$ is the classification accuracy on corruption $c$ at level $s$.

**Models.** The test-time optimization techniques can usually be applied across architectures. We use several classic 2D, 3D CNN-based architectures and transformers as the backbone to show the superiority of our method TeCo. For 2D CNN, we choose vanilla ResNet18 and its variant TAM-ResNet18 (Fan et al., 2019) as the model architecture. The vanilla ResNet18 treats single image frames in the video as the input data. It has only an average fusion at the output stage for the overall video classification. TAM-ResNet18 integrates a temporal module in the ResNet architecture. It helps the model store more temporal information for classification. Hence, it obtains improvements in clean accuracy, especially on the SSV2 dataset. For 3D CNN, we use the 3D version ResNet18 (Hara et al., 2017) for training and evaluation. The 3D convolution layers enable it to capture sufficient spatio-temporal features in an end-to-end training manner. Since TeCo is model agnostic in principle, we use the state-of-the-art transformer MViTv2-S (Li et al., 2022) as the backbone as well. Different from CNN-based architecture, the transformer uses Layer Normalization (Ba et al., 2016).

**Baseline.** Though the video classification area lacks research in test-time optimization, we import the baseline methods in image-based test-time optimization. BN (test-time batch normalization) (Schneider et al., 2020) is a simple method that adapts the batch normalization statistics with test-time data. Tent (Wang et al., 2021a) extends from BN by updating the transformation parameters in the normalization layers, with an objective of entropy minimization. SHOT (Liang et al., 2020) is well-known for combining entropy minimization and pseudo-labeling. TTT (Sun et al., 2020) optimizes the model with a self-supervised auxiliary task at both the training and test stages. Since we only conduct optimization at test stage and have no access to training data in our setting, we denote TTT as TTT*. We choose rotation prediction (Hendrycks et al., 2019) as the self-supervised auxiliary task in TTT*. All the methods can improve the corruption robustness of image-based deep learning models. We re-implement them on the video classification frameworks based on the public available codes.

**Trainig and Evaluation Setup.** We separate the setup of the experiment into three stages: pretraining, test-time optimization, and test-time evaluation. In the pretraining stage, we use uniform sampling to create 16-frame-length input data. Then we use multi-scale cropping on the input for data augmentation and resize the cropped frames to a size of 224. We train the models with an initial learning rate of 0.01 and a cosine annealing learning rate schedule. In test-time optimization, we use the pre-trained model for network weight initialization and update the model parameters for one epoch. For standard (without test-time optimization), BN, Tent, and SHOT methods, we use uniform sampling to extract input from corrupted test data. We apply both uniform and dense sampling to create input for TeCo. In all the test-time optimization experiments, we follow the offline adaptation setting in Wang et al. (2021a). Because the baseline methods have not been implemented in video classification previously, we follow their implementations in image classification and tune the hyper-parameter settings to obtain the best results[1]. For the optimization, we use SGD with momentum. On Mini Kinetics-C, we use a batch size of 32 and a learning rate of 0.001; while for Mini SSV2-C, we use the same batch size but a learning rate of 0.00001. After adapting the models with test data, we freeze the model weights and evaluate on the corrupted data. The inference is only based on the global pathway.

## 4.1 EXPERIMENTAL RESULTS

**mPC across architectures and datasets.** Benefiting from the temporal coherence, our method TeCo consistently outperforms image-based test-time optimization methods across architectures and datasets. Table 1 shows the mPC on Mini Kinetics-C of various methods. Compared with the standard inference, BN improves the mPC by up to 2% because it only adapts BN statistics; Tent, SHOT and TTT* increase the robustness by a larger margin, due to their updates on more parameters. However, the improvements are not consistent across architectures. For TAM-ResNet18, the Tent degrades the performance. We hypothesize that Tent fully removes the training time BN statistics,

---

[1]The detailed implementation details can be found in the appendix.

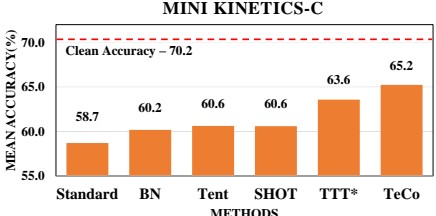 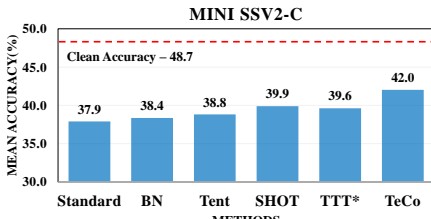

**Figure 2:** Mean Robustness on Mini Kinetics-C and Mini SSV2-C of methods. TeCo reduces the gap between model accuracy on clean and corrupted data significantly.

**Table 1:** mPC across architectures on Mini Kinetics-C and Mini SSV2-C. TeCo outperforms other baseline methods on different architectures and datasets. **Clean Acc** is the accuracy of model tested on clean data.

|  | Backbone | Clean Acc | Standard | BN | Tent | SHOT | TTT* | TeCo |
|---|---|---|---|---|---|---|---|---|
| **Mini Kinetics-C** | 3D ResNet18 | 61.7 | 49.4 | 50.6 | 53.9 | 52.6 | 54.6 | **56.9** |
|  | ResNet18 | 66.0 | 51.6 | 53.0 | 55.6 | 53.2 | 59.5 | **60.8** |
|  | TAM-ResNet18 | 68.5 | 55.9 | 57.1 | 53.8 | 58.4 | 62.2 | **63.4** |
|  | MViTv2-S | 84.4 | 77.9 | 78.0 | 79.2 | 78.2 | 78.0 | **80.1** |
| **Mini SSV2-C** | 3D ResNet18 | 52.2 | 39.3 | 40.0 | 39.9 | 42.4 | 41.5 | **45.7** |
|  | ResNet18 | 30.2 | 20.0 | 20.3 | 22.5 | 23.8 | 22.2 | **24.5** |
|  | TAM-ResNet18 | 55.5 | 44.2 | 45.0 | 44.5 | 45.2 | 46.7 | **49.4** |
|  | MViTv2-S | 56.8 | 48.1 | 48.1 | 48.4 | 48.2 | 48.1 | **48.5** |

but the test-time data is not sufficient to make the parameters reach the global optimum. Though there is a smaller gap between clean and corruption accuracy on MViTv2-S, TeCo is still able to improve the mPC by 2.2%. As a result, our proposed TeCo achieves the best performance, enhancing the mPC by 2.2∼9.2% on various architectures. In Figure 2, we show the mPC on various architectures, including 3D ResNet18, ResNet18, TAM-ResNet18 and MViTv2-S. There is a small gap of 5% between the accuracy testing on clean data and the mPC results achieved by TeCo.

On Mini SSV2-C, our method TeCo surpasses other baseline methods significantly. Table 1 demonstrates the overall mPC of methods on Mini SSV2-C. We find the ResNet18-based method has an obvious degradation in the robustness. Because this method has no module in capturing temporal information except output fusion, their performance on Mini SSV2 is relatively poor when the Mini SSV2 replies more on temporal information for classification. Similar to the phenomenon on Mini-Kinetics-C, BN increases the mPC by up to 1%; Tent, SHOT and TTT* show a diverging enhancement across architectures, up to 3.8%. TeCo consistently improves the corruption robustness on various architectures from 0.4∼6.4%. Horizontally comparing the performance on Mini Kinetics-C and Mini SSV2-C, there is a larger gap between clean accuracy and mPC on Mini SSV2-C from absolute and relative aspects, as shown in Figure 2. We hypothesize that the corrupted temporal information is hard to be compensated, but the spatial semantics can be extracted from the neighboring frames with the nature of temporal coherence.

**Accuracy w.r.t corruption and severity.** When digging deeper into the robustness of methods, we find that TeCo achieves superior performance on corruptions at different levels of severities. Figure 3 shows the complete results on various corruptions from Mini Kinetics-C, with a backbone of 3D ResNet18. The horizontal axis indicates the severity of corruption. When the value is 0, it corresponds to the accuracy on clean test data. We find the Standard and BN methods have much worse performance compared to Tent, SHOT, TTT*, and TeCo, when data is corrupted by shot noise, fog, contrast, saturate change, and rain. We hypothesize that simply adapting the batch normalization statistics is not able to correct the shifts raised by these corruptions. Tent, SHOT, TTT*, and TeCo demonstrate a similar trend when they encounter corruptions at various severity levels. However, TeCo always lies at the top of the trend lines, especially at a severity level of 5. For instance, TeCo maintains the accuracy of 46.7% at level-5 contrast changes, while the accuracy of standard and BN methods drops to 12%. It shows that TeCo benefits from the temporal coherence in video data, apart from an end-to-end test-time optimization scheme. In Appendix, we also show TeCo consistently outperforms other methods on various corruptions on Mini SSV2-C.

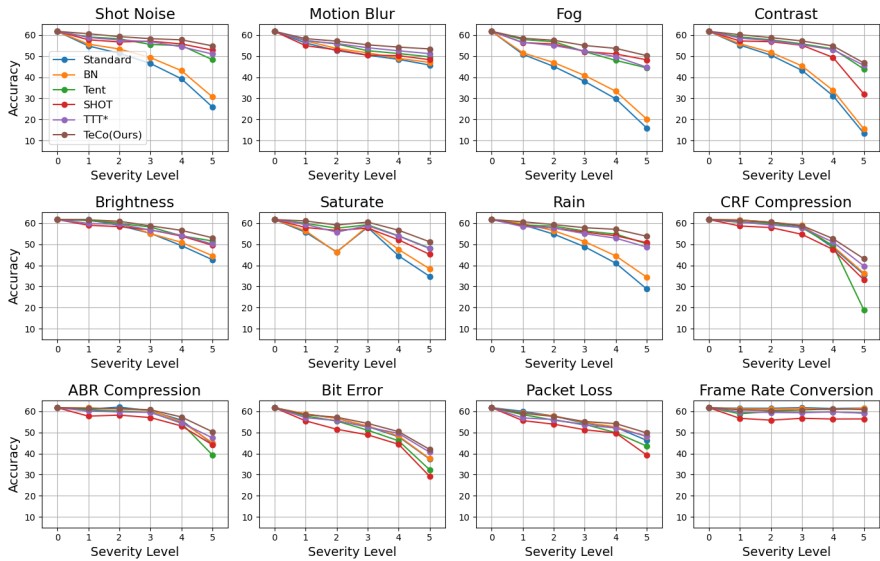

**Figure 3:** Full results on Mini Kinetics-C, with a backbone of 3D ResNet18. The accuracy of standard and BN methods drops sharply when the models encounter severe corruptions like noise, fog, rain, and contrast change. TeCo remains robust against corruptions at even level-5 severity.

**Table 2:** Ablating components of TeCo on Mini Kinetics-C and Mini SSV2-C. The test-time optimization strategy, local pathway (l-path), and attentive temporal coherence regularization improve robustness.

| model, 3D R18 | Standard | SHOT | TeCo (w/o l-path) | TeCo (uniform) | TeCo |
|---|---|---|---|---|---|
| Mini Kinetics-C | 49.4 | 52.6 | 55.9 | 56.5 | **56.9** |
| Mini SSV2-C | 39.3 | 42.4 | 44.4 | 44.5 | **45.7** |

## 4.2 ABLATION STUDIES

**Disentangle Modules in TeCo.** Table 2 shows the utility of three key components in TeCo: test-time optimization strategy, global and local pathways integration, and attentive temporal coherence regularization with local content. The test-time optimization strategy balances training and test-time parameters and statistics, which is more effective than other baseline methods in improving video classification corruption robustness. Global and local pathways provide complementary information for optimization. If we only use uniform sampling for entropy minimization and temporal coherence regularization, the improvement is as low as 0.1% on Mini SSV2-C. When we apply the regularization on local content, the mPC increases by 1% and 1.3% respectively.

**Study Hyper-parameter Sensitivity.** Table 3a demonstrates the impact of $\beta$ in Equation 3. We obtain the best result at $\beta$=1. Table 3b shows the comparison results of the attentive temporal coherence regularization module added after different blocks of 3D ResNet18. The performances after $res_{1-3}$ are similar, while there is a slight drop after $res_4$. We hypothesize the deeper layer will generate more distinguishable features. It benefits from variety instead of consistency for classification. Table 3c studies the impact of batch size on the performance of TeCo. The mPC keeps improving when we increase the batch size from 8 to 32, while it drops when we use the batch size of 64. Because we optimize the model for one epoch for a fair comparison, a larger batch size will lead to fewer iterations of updates. The batch size of 32 balances the amount of data in one batch and the number of iterations in one epoch, which enables models to obtain the best performance. Hence, TeCo can improve robustness reliably without tuning the hyper-parameters carefully.

**Balance training and test statistics with** $\alpha$. Figure 4 shows the mPC on Mini Kinetics-C w.r.t optimization iteration, with different $\alpha$ and the backbone of 3D ResNet18. The $\alpha$ in Section 3.1 controls the weights of training and test statistics in normalization layer initialization. When we set $\alpha$ to a smaller value (e.g., 0.2), the normalization layer relies more on test data initially. Hence, it converges faster in test-time optimization. After 40 iterations of test-time optimization, the mPC is

**Table 3:** TeCo ablation experiments with 3D ResNet18 on Mini Kinetics-C. If not specified, the default setting is: the beta $\beta$ that control the weight of temporal coherence regularization is 1, the regularization is applied after $res_2$ stage of model, the test time optimization uses a batch size of 32 and $\alpha$ is 0.4. The default setting has bold text.

**(a)** $\beta$ **in Equation 3.** TeCo applies a balanced weight to temporal coherence regularization.

| $\beta$ | mPC |
|---|---|
| 0.1 | 56.4 |
| 0.5 | 56.7 |
| **1** | **56.9** |
| 5 | 56.6 |

**(b) Block stage.** The temporal coherence regularization module is added after certain block stage.

| Stage | mPC |
|---|---|
| $res_1$ | 56.6 |
| $res_2$ | **56.9** |
| $res_3$ | 56.6 |
| $res_4$ | 56.2 |

**(c) Batch size.** TeCo uses batches of data in test time optimization.

| Batch Size | mPC |
|---|---|
| 8 | 56.6 |
| 16 | 56.7 |
| **32** | **56.9** |
| 64 | 56.6 |

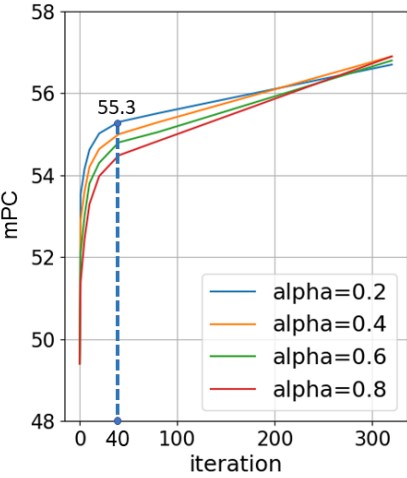



**Figure 4:** Iteration in test-time optimization vs mPC, on various $\alpha$. The mPC improves rapidly in the beginning, then the improvements slow down when more data is used. Smaller $\alpha$ uses more test data statistics, which enables the performance to converge faster.

**Figure 5:** We visualize $res_1$ feature maps of consecutive frames. The time gap between frames is $i = 1$. TeCo generates smoother feature maps in both spatial and temporal dimensions.

enhanced from 49.6% to 55.3%. When the iterations increases, the performance improves slower and gets closer for different $\alpha$. In summary, the model robustness is more sensitive to $\alpha$ when less data is used in test-time optimization.

**Visualize Space-Time Feature Map.** Figure 5 shows the feature maps of noise corrupted video after $res_1$ stages. Horizontally, we visualize the features of different frames. The time gap between neighboring frames is $i = 1$. Standard and BN methods fail to recover human-recognizable semantics from the noise-corrupted frames. Tent obtains noisy outlines along the temporal dimension. TeCo removes the noise and generates temporal coherent feature maps, indicating that TeCo helps the model obtain smoother and more consistent features.

## 5 CONCLUSION

TeCo is a test-time optimization technique for improving the corruption robustness of video classification models, which updates model parameters during testing by two objectives with self-supervised learning: TeCo uses global information for entropy minimization, and it also applies attentive temporal coherence regularization on local information. Benefiting from long-term and short-term spatio-temporal information, TeCo achieves significant corruption robustness improvement on Mini Kinetics-C and Mini SSV2-C. We hope TeCo becomes a new baseline method that makes video classification models more reliable and robust in real-world deployment.

## 6 ACKNOWLEDGEMENTS

This work was done at Rapid-Rich Object Search (ROSE) Lab, Nanyang Technological University. This research is supported in part by the NTU-PKU Joint Research Institute (a collaboration between the Nanyang Technological University and Peking University that is sponsored by a donation from the Ng Teng Fong Charitable Foundation). This work is also supported by the Research Grant Council(RGC) of Hong Kong through Early Career Scheme (ECS) under the Grant21200522,CityU Applied Research Grant(ARG) 9667244,and Sichuan Sci-ence and Technology Program 2022NS-FSC0551.

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

# A   APPENDIX

## A.1   SOURCE-FREE DOMAIN ADAPTATION

We benchmark the video domain adaptation for shift from UCF to HMDB and vice versa. In our setting, we have only access to test data and pre-trained model. In the domain adaptation problem, 12 common classes of 3209 video data in UCF and HMDB are used, following the official split provided by (Chen et al., 2019). For the pre-trained models, we use Kinetics-pretrained 3D ResNet50. Then we fine tune the model with training data in UCF and HMDB.

Table 4 reports the target accuracy of standard and test-time optimization methods. We denote Tent as Tent* because we remain the training statistics in normalization layer, otherwise the performance will drop significantly. All the methods can help model adapt to video data with domain shift. TTT* obtains competitive improvement of 2.7% on UCF-HMDB. Tent* also increases accuracy on HMDB-UCF by 2.4%. We find that TeCo consistently outperforms other baseline methods. When we train model with UCF and optimize on HMDB at test time, it improves the accuracy by 3.3%. Similarly, on HMDB-UCF, the accuracy is enhanced by 2.8%.

**Table 4:** Adapting models to target domain without access to source data. We use Kinetics for model pre-training, and fine tune the model with UCF and HMDB training data respectively. UCF-HMDB means training on UCF and testing on HMDB data.

| model, 3D R50 | Standard | BN | Tent* | SHOT | TTT* | TeCo |
|---|---|---|---|---|---|---|
| UCF-HMDB | 74.2 | 75.6 | 76.4 | 76.7 | 76.9 | **77.5** |
| HMDB-UCF | 80.7 | 82.6 | 83.1 | 82.7 | 81.9 | **83.5** |

## A.2   FULL RESULTS ON MINI SSV2-C

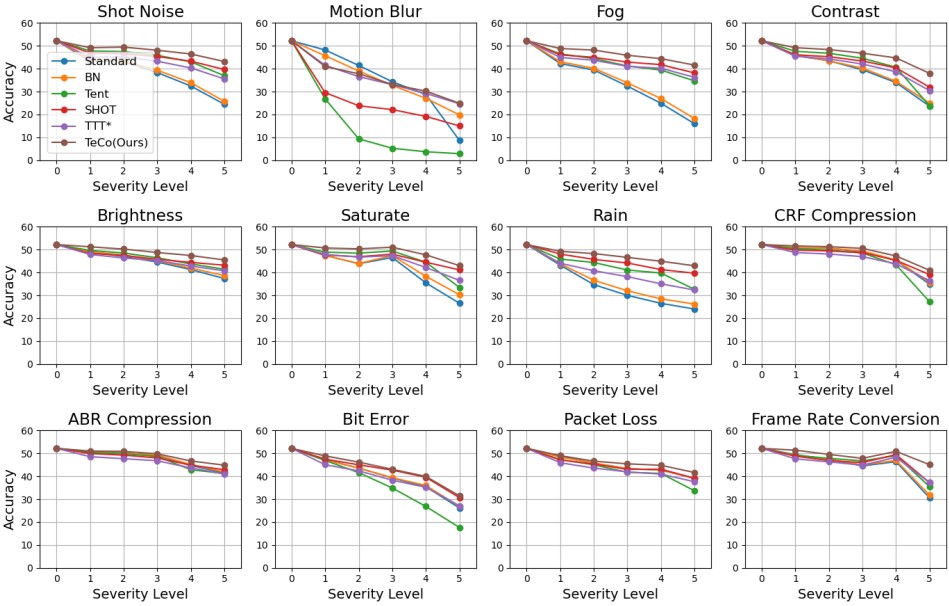

**Figure 6:** Full results on Mini SSV2-C, with a backbone of 3D ResNet18. Tent becomes vulnerable to motion blur and bit error, while TeCo lies above other methods on most types of corruptions.

Figure 6 shows the complete results on Mini SSV2-C. Apart from the corruptions which cause severe accuracy degradation in Mini Kinetics-C, the corruptions like motion blur, bit error, and frame rate conversion also lead to significant drops in accuracy in Mini SSV2-C. Intuitively, these three types

of corruption have more impact on temporal information. For example, bit error propagates the perturbation in the consecutive frames, and frame rate conversion changes the speed of motions in the video clips. The performance of Tent is devastating when facing motion blur and bit error. In contrast, TeCo, which uses a balanced test-time optimization framework, guarantees performance even though the test-time data has a large shift from the clean data. As a result, TeCo achieves consistent robustness improvements on different corruptions across datasets.

## A.3    Baseline Implementation and Hyper-parameters

Because all the baseline methods are model-agnostic, we follow the implementation of the original papers and code. BN simply controls the weights of training and test statistics in the normalization layers. We denote the hyper-parameter as $\alpha$ in Section 3.1 as well. Tent and TTT* have no specific hyper-parameter, but they are sensitive to the learning rate. SHOT contains two objectives. One is pseudo-label classification loss and the other is entropy loss. We use $\lambda$ to indicate the weight of the pseudo-label classification loss. For a fair comparison, we use a batch size of 32 in all the implementations.

**Table 5:** Implementation detail for baseline methods

| Dataset | Baseline | Hyper-Parameter | Learning Rate |
|---|---|---|---|
| Mini Kinetics-C | BN | weight of training statistics $\alpha$=0.5 | - |
| | Tent | - | 1e-3 |
| | SHOT | weight of pseudo-label classification loss $\lambda$=0.3 | 1e-3 |
| | TTT* | - | 1e-4 |
| Mini SSV2-C | BN | weight of training statistics $\alpha$=0.6 | - |
| | Tent | - | 1e-5 |
| | SHOT | weight of pseudo-label classification loss $\lambda = 0.3$ | 1e-5 |
| | TTT* | - | 1e-5 |

## A.4    Ablating l1/l2 Distance and Partial/Full Parameter Update

In the default setting of TeCo, we use $l_1$ distance to measure the similarity of features along the time dimension. The $l1$ loss will penalize small perturbations and encourage sparsity in the feature maps, which enhances model robustness (Guo et al., 2018; Chen et al., 2022). $l2$ loss is also widely used in de-noising. It usually leads to smoother change. We find that $l2$ distance also achieves promising performance on corruption robustness. We obtain a mPC of 56.8% when TeCo follows the default settings. It remains an open question to explore the optimal metric for measuring similarity.

We also conduct an ablation study on Mini Kinetics-C and UCF-HMDB datasets to understand the impact of network parameter updates. TeCo updates part of network parameters as mentioned in Section 3.1. Table 6 shows that fully updating network parameters may hurt the performance. Especially for small datasets like UCF-HMDB, the accuracy drops by 1.1% on UCF-HMDB and by 1.3% HMDB-UCF. Limiting the parameters which can be updated can improve both optimization stability and efficiency (Wang et al., 2021a).

**Table 6: Partially/Fully update network parameters**. Comparison of experiments with updating all/partial network parameters. All the experiments in this table followed the default hyper-parameters.

| Network Parameters | Mini Kinetics-C | UCF-HMDB | HMDB-UCF |
|---|---|---|---|
| Partially | **56.9** | **77.5** | **83.5** |
| Fully | 56.2(-0.7) | 76.4(-1.1) | 82.2(-1.3) |

