# OpenReview forum: "Temporal Coherent Test Time Optimization for Robust Video Classification"
_ICLR.cc/2023/Conference — ICLR 2023 poster_

### Official Review · Reviewer_bEhU · 2022-10-20

**Confidence:** 4
**Correctness:** 3
**Technical Novelty And Significance:** 3
**Empirical Novelty And Significance:** 3
**Recommendation:** 6

**Clarity, Quality, Novelty And Reproducibility:**

The paper writing is good and implementation details are specific, which benefits reproducibility.

**Strength And Weaknesses:**

Strength:
+ The two regularization is neat and makes sense. Especially, the attention module to compute the temporal affinity smooths the feature and constrains the temporal coherence, which implements the motivation precisely.
+ The method achieves strong results and conducts a thorough ablation study.
+ The paper writing is good and easy to understand.

Weakness:
+ Figure 3 and Figure 4 are surplus. Leaving one figure in the main paper would be enough.
+ The ablation study of tuning the whole parameters of the model is missing. Otherwise, only tuning shallow layers and BN layers in the deep layers sounds unreasonable to me. It would be better to provide some experimental evidence.

**Summary Of The Paper:**

The paper proposes a method of test-time adaptation for video action classification. In specific, the paper introduces two regularization losses. One is the temporal coherence regularization on the local pathway to guarantee the smoothness of adjacent frames, while the other is the entropy minimization to enhance the robustness of the shallow layers. The experiments show that the proposed method achieves the best results on two corruption robustness datasets (Mini Kinetics-C, Mini SSV2-C).


**Summary Of The Review:**

Overall, it is a borderline paper with clear motivation and a reasonable method. The downside is the experimental analysis, which could be more informative and highlight the contribution.

---

> ### Author Response · Authors · 2022-11-15
> **Thank you for your valuable comments. Please find our initial responses below.**
>
> **Q1:** *Figure 3 and Figure 4 are surplus. Leaving one figure in the main paper would be enough.*
>
> **Ans:** We shift Figure 4 to Appendix and show the full results of Mini SSV2 in Figure 6 in Appendix.
>
>
> **Q2:** *The ablation study of tuning the whole parameters of the model is missing. Otherwise, only tuning shallow layers and BN layers in the deep layers sounds unreasonable to me. It would be better to provide some experimental evidence.*
>
> **Ans:** We added an ablation study on partially/fully updating parameters in Appendix A.4. The results show that tuning the whole parameters will hurt the performance. The degradation of performance is worse on smaller dataset and larger domain shift, because tuning the whole parameters may reduce the stability of test-time optimization[1]. We show the results on Mini Kinetics-C and UCF-HMDB below:
>
> | Network Parameter | Mini Kinetics-C | UCF-HMDB| HMDB-UCF |
> | ------------- | ------------- |------------- | ------------- |
> | Partially update | 56.9  | 77.5 | 83.5
> | Fully update  | 56.2(-0.7) | 76.4(-1.1) | 82.2(-1.3)
>
> [1]Wang, Dequan, et al. "Tent: Fully test-time adaptation by entropy minimization." ICLR (2021).

---

> > ### Comment · Reviewer_bEhU · 2022-11-17
> > **Response to Rebuttal**
> >
> > Thank the authors for their response. After reading other reviews and the modified paper, I think it is a good paper for the community. Thus, I keep my original rating of acceptance.

---

> > > ### Author Response · Authors · 2022-11-17
> > > **Thank you for the response**
> > >
> > > Thank you so much for your response! We hope the paper can help the community as well.

---

### Official Review · Reviewer_rshF · 2022-10-24

**Confidence:** 4
**Correctness:** 3
**Technical Novelty And Significance:** 2
**Empirical Novelty And Significance:** 2
**Recommendation:** 6

**Clarity, Quality, Novelty And Reproducibility:**

Clarity is good. I can understand the proposed method easily.
The novelty seems on the weaker side to me, because temporal consistency is not new (e.g., Azimi et al. also uses some sort of temporal consistency).
The experiments are not entirely convincing (pls see discussions above).

**Strength And Weaknesses:**

Strength:
+ Advancing TTT can potentially have a big impact on real-world computer vision applications.
+ The authors evaluate TeCo on more than one dataset, and demonstrate consistent improvement on both.

Weakness:
- The datasets are not commonly used ones in video research or applications. Mini Kinetic-C and Mini SSV2-C are small and the corruptions are artificially generated. Regarding the size, the larger the pretraining dataset is, the less important TTT could be. Thus, instead of experimenting on small scales, it might be more realistic to understand the methods on commonly used larger datasets (e.g. full Kinetics or full SSv2).
Regarding the corruptions, I'm not convinced that dealing with these artificially generated corruptions are a great way to study generalization. A better way can be to use full Kinetics and SSV2 to pre-train the model, and then use some "real-world datasets on other domains, potentially with corruptions" to evaluate the TTT. After all, if one cannot easily find video datasets with "corruptions", maybe that suggests the setting isn't very practically useful.

- The models used are small, proposed in ~2017-2019, and perform significantly worse than todays' state-of-the-art. Video architectures have made significant improvements over the last 3-5 years, so one might observe different observations with newers models (e.g. Swin, MViT, etc.).

- There are a number of video TTT models such as, Azimi et al. 2022, but they are not compared.

- The authors note that "For standard (without test-time optimization), BN, Tent, and SHOT methods, we use
uniform sampling to extract input from corrupted test data. We apply both uniform and dense sampling to create input for TeCo.". I wonder if using "dense sampling" contributes the higher performance (because effectively the model sees more diverse inputs and can be seen as a type of data augmentation. )

- I wonder how the baselines are implemented and how are the hyperparameters picked. Since the original baseline papers didn't perform these experiments, I assume the authors implemented these experiments. I wonder if the hyperparameters picked would affect the results.


**Summary Of The Paper:**

In this paper, the authors propose a test-time training (TTT) method called TeCo, which uses not only a standard classification loss, but also a temporal consistency loss, for TTT on video data. On Mini Kinetic-C and Mini SSV2-C, the authors compared TeCo with a number of baseline methods and show that TeCo outperforms prior work on the evaluated settings.

**Summary Of The Review:**

Overall, my main concern is regarding experiments. I find the experiment settings limited, and I'm not convinced by the experiments that the proposed method is effective. A more realistic setting (including real datasets, stronger models) would have been more convincing.


--- post rebuttal ---
Thanks for providing the rebuttal. The additional experiments comparing with TTT* and the experiments using MViT are helpful and clarify many of my concerns. I also find the domain shift experiments on HMDB and UCF101 very nice and convincing. Regarding the dataset scale, my concerns are still not fully addressed. One potential way to understand the effect of pretraining data scale is to evaluate the proposed method under increasingly larger "pretraining dataset size", to see if the improvement still holds when pre-training dataset is large.

Overall, with the new results, I find the paper greatly improved, and I adjusted the rating accordingly.

---

> ### Author Response · Authors · 2022-11-15
> **Thank you for your comments and feedbacks. We added experiments with more settings.(Part1)**
>
> Thank you for your valuable comments. We would like to clarify that our method uses two self-supervised learning objectives because no label information is used: one is entropy minimization, which is different from standard classification loss (e.g., cross-entropy loss); the other is temporal coherence regularization. Please find our initial responses for the comments below:
>
> **Q1:** *The datasets are not commonly used ones in video research or applications. Mini Kinetic-C and Mini SSV2-C are small and the corruptions are artificially generated. Regarding the size, the larger the pretraining dataset is, the less important TTT could be. Thus, instead of experimenting on small scales, it might be more realistic to understand the methods on commonly used larger datasets (e.g. full Kinetics or full SSv2). Regarding the corruptions, I'm not convinced that dealing with these artificially generated corruptions are a great way to study generalization. A better way can be to use full Kinetics and SSV2 to pre-train the model, and then use some "real-world datasets on other domains, potentially with corruptions" to evaluate the TTT. After all, if one cannot easily find video datasets with "corruptions", maybe that suggests the setting isn't very practically useful.*
>
> **Ans:** Mini Kinetics and Mini SSV2 have around half size of their original datasets, so the test dataset of Mini Kinetics has around 10K videos and the Mini SSV2 test dataset contains around 13K videos. They are used in the large-scale empirical analysis of video data recently [1][2][3].
>
> We follow [2] to use 12 types of corruption (each corruption consists of 5 levels of severity). Most types of corruption are considered in image corruption robustness benchmarks [4] as well, and they are widely adopted in robust generalization studies among the community.
>
> Furthermore, we agree that the pretraining model with the full dataset and test on real-world datasets with domain shift makes our method more convincing. Hence, we pretrain 3D ResNet50 with the full Kinetics dataset, fine-tune the pre-trained model with the UCF-HMDB dataset, and evaluate our method on the shift from UCF to HMDB and vice versa. We show the results in Appendix Table 3. It demonstrates that the baseline methods can improve the accuracy in domain adaption. Our method TeCo also obtains promising performance though we focus on corruption robustness. We show the results on UCF-HMDB below:
>
> | model, 3D R50 | Standard | BN | Tent* | SHOT| TTT*|TeCo |
> | ------------- | ------------- |------------- | ------------- |------------- | ------------- |------------- |
> | UCF-HMDB| 74.2  | 75.6 | 76.4 | 76.7| 76.9|**77.5**|
> | HMDB-UCF | 80.7 | 82.6 | 83.1 | 82.7 | 81.9 | **83.5**|
>
> **Q2:** *The models used are small, proposed in ~2017-2019, and perform significantly worse than todays' state-of-the-art. Video architectures have made significant improvements over the last 3-5 years, so one might observe different observations with newers models (e.g. Swin, MViT, etc.).*
>
> **Ans:** We added the transformer-based model MViTv2(Li Y et al. 2022) in our experiments. We show that MViTv2 achieves significant robustness improvement on Mini Kinetics-C, and TeCo can further improve the model performance on both datasets. We have added the MViTv2-based results in Table 1 of the revised manuscript.
>
> |Dataset | Backbone| Clean Acc| Standard | BN | Tent | SHOT| TTT*|TeCo |
> | ------------- | --------- |-------- | --------- |------------- | ------------- |------------- | ------------- |------------- |
> | Mini Kinetics-C| MViTv2-S | 84.4 | 77.9 | 78.0| 79.2|78.2| 78.0|**80.1**|
> | Mini SSV2-C | MViTv2-S | 56.8 | 48.1 | 48.1 | 48.4 | 48.2|48.1|**48.5**|
>
>
> [1]Chen, Chun-Fu Richard, et al. "Deep analysis of cnn-based spatio-temporal representations for action recognition." Proceedings of the IEEE/CVF Conference on Computer Vision and Pattern Recognition. 2021.
>
> [2]Yi, Chenyu, et al. "Benchmarking the Robustness of Spatial-Temporal Models Against Corruptions." Advances in neural information processing systems. 2021.
>
> [3]Singh, Ankit, et al. "Semi-supervised action recognition with temporal contrastive learning." Proceedings of the IEEE/CVF Conference on Computer Vision and Pattern Recognition. 2021.
>
> [4] Hendrycks, Dan, and Thomas Dietterich. "Benchmarking neural network robustness to common corruptions and perturbations." ICLR. 2019.

---

> ### Author Response · Authors · 2022-11-15
> **Thank you for your comments and feedbacks. We added experiments with more settings. (Part 2)**
>
> **Q3:** *There are a number of video TTT models such as, Azimi et al. 2022, but they are not compared.*
>
> **Ans:** We added TTT* as a baseline method in our comparion. The experimental results are shown in Table 1 of the revised vision. TTT(Sun, Yu, et al. 2020) and its variants jointly train the model with supervised loss and auxiliary loss at training time, then they optimize the model at test time. However, we follow the setting of Tent and assume that training data is not accessible, so we denote TTT as TTT*.
>
> Since Azimi et al. 2022 used self-supervised models of dense tracking under the setting of TTT, it is not applicable to the classification task. We use rotation prediction in TTT(Sun, Yu, et al. 2020) as the auxiliary task and optimize the pre-trained model with test video data. We show the results below:
>
> | Dataset | Backbone | Standard| TTT*| TeCo|
> | ------------- | ------------- |------------- | ------------- |------------- |
> |Mini Kinetics-C| 3D ResNet18  | 49.4 | 54.6 |56.9|
> |                         | ResNet18  | 51.6 | 59.5 |60.8|
> |                         | TAM-ResNet18  | 55.9 | 62.2 |63.4|
> |                         | MViTv2-S  | 77.9 | 78.0 |80.1|
> |Mini SSV2-C| 3D ResNet18  | 39.3 | 41.5 |45.7|
> |                         | ResNet18  | 20.0 | 22.2 |24.5|
> |                         | TAM-ResNet18  |44.2 | 46.7 |49.4|
> |                         | MViTv2-S  |48.1 | 48.1 |48.5|
>
> **Q4:** *The authors note that "For standard (without test-time optimization), BN, Tent, and SHOT methods, we use uniform sampling to extract input from corrupted test data. We apply both uniform and dense sampling to create input for TeCo.". I wonder if using "dense sampling" contributes the higher performance (because effectively the model sees more diverse inputs and can be seen as a type of data augmentation. )*
>
> **Ans:** In the original manuscript, we conducted an ablation study to disentangle the effectiveness of modules in TeCo. In Table 2, 'TeCo(Uniform)' only uses uniform sampling to obtain input data. In comparison, 'TeCo' improves the mPC by 0.4% and 1.2% on the two datasets when using both uniform and dense sampling. Simply speaking, the dense sampling does contribute to higher performance. It explains the motivation that we construct the method with global and local pathways.
>
> **Q5:** *I wonder how the baselines are implemented and how are the hyperparameters picked. Since the original baseline papers didn't perform these experiments, I assume the authors implemented these experiments. I wonder if the hyperparameters picked would affect the results.*
>
> **Ans:** All the baseline methods are model agnostic, and the papers have publicly available code. We implement the baseline methods based on the paper and code provided. In the implementation, we find the methods are sensitive to several hyper-parameters. For example, BN is sensitive to $\alpha$; Tent is sensitive to the learning rate. Therefore, we tune the hyper-parameters and report the best results in our paper. Besides, we provide hyper-parameters of the baseline methods for both datasets in Appendix Table 5.
>
> **Q6:** *The novelty seems on the weaker side to me, because temporal consistency is not new (e.g., Azimi et al. also uses some sort of temporal consistency).*
>
> **Ans:**  Azimi et al. 2022 combines existing self-supervised models and TTT to alleviate covariate shifts in dense tracking problems.
> Our rationality and motivation differ in various aspects:
> 1. The temporal consistency/coherence is under-explored in the robust video classification.
> 2. The corruptions and perturbations in the consecutive frames will cause inconsistency.
> 3. Different parts of video frames may change at different speeds along the temporal dimension (e.g., the background changes slower). Motivated by this phenomenon,  we use an attentive module to encourage more consistency in the background/relatively static area.
>
> As a result, we design a novel framework instead of using existing self-supervised models for optimization.

---

> > ### Comment · Reviewer_rshF · 2022-11-22
> > **Thanks for providing the rebuttal**
> >
> > Thanks for providing the rebuttal. Please see my updated review above.

---

> > > ### Author Response · Authors · 2022-12-02
> > > **Thank you for your response.**
> > >
> > > Thank you for the valuable feedback! It helps improve our work significantly. The effect of pre-training dataset scale in test-time adaption could be an interesting topic to explore. We will study it more comprehensively in both image and video domains in future work.

---

### Official Review · Reviewer_naa6 · 2022-11-02

**Confidence:** 2
**Correctness:** 4
**Technical Novelty And Significance:** 3
**Empirical Novelty And Significance:** 3
**Recommendation:** 6

**Clarity, Quality, Novelty And Reproducibility:**

In general, the exposition was not always clear, some parts of the paper were hard to follow. The authors introduce hyperparameters but never explain them, or even state how they are set during training, which makes it impossible to reproduce the results in the paper.

* The authors use several symbols without introducing them. For example, what are the g and f functions in Section 3.2? They also appear in Figure 1, but are never introduced in the text.

* The hyperparemters alpha and beta are introduced in the beginning Section 3.1, but it’s never discussed how these variables are set during the experiments, or how sensitive the method is to changing this variable.  There is (an apparently different) beta turning up in Eq. 3.4. Again, there is no discussion about how to set this hyperparameter or how sensitive results are to changing this.

*  Eq (1): Why L1 distance? Wouldn’t L2 distance make more sense? (i.e., do not penalize small pertubations?)


Minor remarks:
We propose to build our method upon the test-time optimization which updates all the parameters in shallow layers and only normalization layers in the deep layers. => What are "shallow layers" and what are "deep layers"? From Figure 1, I think you mean the initial (lower) layers are shallow and the layers deeper in the network are the "deep layers". But this formulation is ambiguous and not clearly defined.

“TeCo increases 8% and 5.4% average accuracy across backbones on Mini Kinetics-C and Mini SSV2-C” => TeCo increases accuracy by 8% on across backbones on Mini Kinetics-C and by 5.4% on Mini SSV2-C.

“Recently there emerge studies” => Recently, studies have emerged

“they have subtle enhancement on video data” => they only show minor improvements on video data



**Strength And Weaknesses:**

Strenghts:
* The method seems applicable to any video model
* The empirical results look good

Weaknesses:
* The method is not evaluated on Transformer-based models. It would be nice (though not necessary) if this would be added.
* The authors do not discuss runtime implications: how expensive is it to run this method? How long do you need to tune this? How much data do you require to obtain reasonable results? It would be nice if these points could be adressed.
* There are no explanations/ablations on some of the hyperparameters that the metod introduced (See "Clarity" below)
* The authors need to work on Clarity/Reproducibility (see further down in the review)

**Summary Of The Paper:**

The authors propose a method that deal with input domain shifts in videos during test time (I.e., it deals with corruptions not observed in training data, like different weather conditions).  They propose two self-supervised objectives that are applied to the test-data: one minimizes the entropy of predictions across frames, the other penalizes features/representations that change too strongly during different frames of the video. Results on Mini Kinetics C and Mini SSV2-C are provided.

**Summary Of The Review:**

The authors introduce a method to optimize a network for test-time domain shift. Empirical results look promising, but in its current state the authors need to adress some clarity and reproducibility concerns.

UPDATE: After seeing the updated version by the authors, I feel like most of my weak points have been adequately addressed. I'm not familiar enough with the problem it is addressing to judge it's relevance to the community. But both motivation and empirical validation seem sound to me, so I would like to raise my rating to a 7 (i.e., "good paper, but unclear about practical relevance").

---

> ### Author Response · Authors · 2022-11-15
> **Thank you for your comments and feedback. Please find our initial response below.**
>
> **Q1:**  *The method is not evaluated on Transformer-based models. It would be nice (though not necessary) if this would be added.*
>
> **Ans:** Thank you for your valuable feedback! We added MViTv2[1] as a backbone in Table 1 in the revised manuscript. As a model-agnostic method, TeCo can consistently improve the corruption robustness of the transformer-based video classification model. The results are shown below:
>
> | Dataset   | Backbone | Clean Acc| Standard | BN | Tent |      SHOT|    TTT*         | TeCo
> | ----------- | ----------- | ----------- | ----------- | ----------- | ----------- | ----------- | ----------- | ----------- |
> | Mini Kinetics-C| MViTv2-S  |84.4 | 77.9| 78.0| 79.2| 78.2| 78.0| **80.1**|
> | Mini SSV2-C    | MViTv2-S | 56.8| 48.1 | 48.1| 48.4| 48.2|48.1 |**48.5**|
>
> **Q2:** *The authors do not discuss runtime implications: how expensive is it to run this method? How long do you need to tune this? How much data do you require to obtain reasonable results? It would be nice if these points could be addressed.*
>
> **Ans:** For all the methods, we run the test-time optimization for one epoch and use a batch size of 32. Since Mini Kinetics has around 10k videos in the test dataset, we will run ~300 iterations to obtain the final results. We added an ablation study to demonstrate the impact of iterations in our method. Figure 4 (in the revised manuscript) shows the mPC vs Iterations with different $\alpha$. When we set $\alpha$ to 0.2, the mPC is improved from \%49.6 to \%55.3 with only 40 iterations (1280 videos). When more data is used, the performance improvement slows down.
>
>
> **Q3:** *The authors use several symbols without introducing them. For example, what are the g and f functions in Section 3.2? They also appear in Figure 1, but are never introduced in the text.*
>
> **Ans:** f($\cdot$) stands for the shallow layer and g($\cdot$) represents the deep layers. We have added the explanation of symbols in the revised manuscript.
>
>
> **Q4:** *The hyperparameters alpha and beta are introduced at the beginning Section 3.1, but it’s never discussed how these variables are set during the experiments, or how sensitive the method is to change this variable. There is (an apparently different) beta turning up in Eq. 3.4. Again, there is no discussion about how to set this hyperparameter or how sensitive results are to changing this.*
>
> **Ans:** We added ablation studies on $\alpha$ and $\beta$ in Figure 4 and Table 3 (in the revised version). It shows that the performance is sensitive to $\alpha$ when less data is used. However, when the data amount increases, the performance is not highly sensitive to the hyper-parameters $\alpha$ and $\beta$. To improve the reproducibility, we provide the default setting in the caption of Table 3. We show the ablation of $\beta$ below:
>
>
> | **$\beta$**  | 0.1| 0.5 | 1 |     5
> | ----------- | ----------- | ----------- | ----------- | -----------
> | **mPC** | 56.4 | 56.7 | **56.9** | 56.6
>
>
> **Q5:** *Eq (1): Why L1 distance? Wouldn’t L2 distance make more sense? (i.e., do not penalize small perturbations?)*
>
> **Ans:** L1 norm puts more penalty on outliers, which makes models more robust [2][3]. We also agree that L2 distance is a widely used metric in regularization, which leads to smooth changes. With a simple ablation, we found that L2 distance also obtains a promising performance of \%56.8.
>
> **Q6:** *The exposition is not clear enough and some parts need clarification.*
>
> **Ans:** We have explained the symbols g, h, and f, and further clarified the definition of shallow and deep layers in the Method section. We also modified the writing based on your advice.
>
> [1] Li, Yanghao, et al. "MViTv2: Improved Multiscale Vision Transformers for Classification and Detection." Proceedings of the IEEE/CVF Conference on Computer Vision and Pattern Recognition. 2022.
>
> [2] Alizadeh, Milad, et al. "Gradient $\ell_1 $ regularization for quantization robustness." ICLR(2020).
>
> [3] Bektaş, Sebahattin, and Yasemin Şişman. "The comparison of L1 and L2-norm minimization methods." International Journal of the Physical Sciences 5.11 (2010): 1721-1727.

---

> > ### Comment · Reviewer_naa6 · 2022-11-15
> > **Response to Rebuttal**
> >
> > Thank you for your clarifications and for the additional experiments, they are highly appreciated. I feel like adding MViT especially adds value to the paper by showing the impact on modern methods.
> >
> > It seems to me like TeCo is a solid contribution, and I will update my review score accordingly.

---

> > > ### Author Response · Authors · 2022-11-15
> > > **Thank you for your response**
> > >
> > > Thank you so much for your encouraging responses. We also wish TeCo can benefit the video analysis and robustness community.

---

### Author Response · Authors · 2022-11-15
**Summary of modification**

We thank all reviewers for the insightful comments to improve our work, we label the modification in red color in the revised manuscript and summarize the updates as below:
1. We added a transformer-based backbone MViTv2 (Li Y et al. 2022) in our experiments and show the results in Table 1.
2. We added a baseline method TTT* for comparison. We show the results in Table 1.
3. We added an ablation study on hyper-parameters $\beta$ in Table 3. We present the default setting for Mini Kinetics-C in the caption of Table 3.
4. We added an ablation study on $\alpha$ and optimization iterations in Figure 4.
5. We added experiments of UCF-HMDB with 3D ResNet50 in Appendix Section A.1.
6. We added baseline method implementation details in Appendix Section A.3.
7. We added ablation studies on l1/l2 distance and partially/fully updating parameters in Appendix Section A.4.
8. We clarified the symbols and definitions in the Method chapter.
9. We modified the experimental analysis based on updated results. We moved the full results of Mini SSV2-C to Appendix Section A.2

---

### Decision · Program_Chairs · 2023-01-20

**Decision:**

Accept: poster

**Justification For Why Not Higher Score:**

- Main datasets are not commonly used in video classification (artificially generated Mini-Kinetics and Mini-SSv2), relatively small dataset sizes

**Justification For Why Not Lower Score:**

- Important problem with real-world applications
- Thorough empirical evaluation (including the rebuttal phase)

**Metareview: Summary, Strengths And Weaknesses:**

Test-time optimisation is an effective way to increase model robustness to corrupted data without having to retrain the model. While the problem was studied in the context of images, it received less attention in the context of videos. In this work the authors propose a method to improve the test-time robustness to distribution shifts in video classification models. In particular, two self-supervised objectives are coupled with the standard classification loss, namely entropy minimisation and temporal coherence/consistency regularisation. The results are evaluated on two corrupted datasets (Mini Kinetics-C, mini SSv2-C) where the method outperforms prior work.

The reviewers appreciated the clarity of writing, the significance of the problem, and the fact that the suggested regularisers match the motivation laid out by the authors. The rebuttal did a good job convincing the reviewers in the empirical results. I will recommend the acceptance of this work. Please add all the clarifications and ablations from the rebuttal in the revised submission as they played a critical role during the discussion phase.

**Note From Pc:**

if the above contains the word "oral" or "spotlight" please see: "oral" presentation means -> notable-top-5% and "spotlight" means -> notable-top-25%. As stated in our emails, we are disassociating presentation type from AC recommendations